# Change in Blood Flow Velocity Pulse Waveform during Plateau Waves of Intracranial Pressure

**DOI:** 10.3390/brainsci11081000

**Published:** 2021-07-29

**Authors:** Karol Sawicki, Michał M. Placek, Tomasz Łysoń, Zenon Mariak, Robert Chrzanowski, Marek Czosnyka

**Affiliations:** 1Department of Neurosurgery, Medical University of Bialystok, 15-089 Białystok, Poland; tomasz.lyson@umb.edu.pl (T.Ł.); zenon.mariak@umb.edu.pl (Z.M.); robert.e.chrzanowski@gmail.com (R.C.); 2Brain Physics Laboratory, Division of Neurosurgery, Department of Clinical Neurosciences, University of Cambridge, Cambridge CB2 0QQ, UK; mp963@cam.ac.uk (M.M.P.); mc141@medschl.cam.ac.uk (M.C.); 3Department of Biomedical Engineering, Faculty of Fundamental Problems of Technology, Wrocław University of Science and Technology, 50-370 Wrocław, Poland; 4Institute of Electronic Systems, Warsaw University of Technology, 00-661 Warsaw, Poland

**Keywords:** intracranial pressure, transcranial Doppler, non-invasive plateau waves detection

## Abstract

A reliable method for non-invasive detection of dangerous intracranial pressure (ICP) elevations is still unavailable. In this preliminary study, we investigate quantitatively our observation that superimposing waveforms of transcranial Doppler blood flow velocity (FV) and arterial blood pressure (ABP) may help in non-invasive identification of ICP plateau waves. Recordings of FV, ABP and ICP in 160 patients with severe head injury (treated in the Neurocritical Care Unit at Addenbrookes Hospital, Cambridge, UK) were reviewed retrospectively. From that cohort, we identified 18 plateau waves registered in eight patients. A “measure of dissimilarity” (Dissimilarity/Difference Index, DI) between ABP and FV waveforms was calculated in three following steps: 1. fragmentation of ABP and FV signal according to cardiac cycle; 2. obtaining the normalised representative ABP and FV cycles; and finally; 3. assessing their difference, represented by the area between both curves. DI appeared to discriminate ICP plateau waves from baseline episodes slightly better than conventional pulsatility index did: area under ROC curve 0.92 vs. 0.90, sensitivity 0.81 vs. 0.69, accuracy 0.88 vs. 0.84, respectively. The concept of DI, if further tested and improved, might be used for non-invasive detection of ICP plateau waves.

## 1. Introduction

Given the multitude of biological events influencing intracranial pressure (ICP), it should be perceived as “more than a number” [1]. Therefore, using only current ICP value, without secondary analysis of signal and its time-dependent dynamics, contributes to the therapy rather weakly [2,3]. In clinical practice, identifying patterns of ICP changes over time, the best known of which are Lundberg’s “plateau waves” [4], has been proved relevant [5,6,7]. Direct ICP measurement is considered essential in neurocritical care, but it is not commonly used outside this environment due to limitations inherent to its invasive nature. Considering this limitation of “traditional” ICP measurement, in our work, we attempted to explore the potential of TCD in non-invasive detection of plateau waves. Our method consists of superimposing the signal of transcranial Doppler sonography (TCD) on the signal of arterial blood pressure (ABP) before and during plateau waves, and extracting the difference in their shapes.

Plateau waves are observed in up to 25% of patients requiring neurocritical care [8] and manifest as a rapid rise in ICP lasting from a few minutes to a few hours [9]. According to the widely accepted theory presented by Rosner in 1982 [10], the plateau wave is a result of vasodilatory cascade building up on physiological autoregulatory mechanisms in the face of drop in cerebral perfusion. The cost of this reaction is an increase in cerebral blood volume which transfers directly to ICP, thus building a vicious circle. An occurrence of a plateau wave is by no means a benign incident, even though most cases are clinically occult and may cease spontaneously. In some unpredictable instances, the vasodilatory reactions can break up out of “safe” limits to cause refractory brain ischemia and death. For this reason, detection of these spontaneous ICP increases is of utmost clinical significance, while non-invasive detection could be even more beneficial, especially for patients not submitted to invasive ICP monitoring.

Blood flow velocity (FV) measured with transcranial Doppler (TCD) has been widely accepted as a surrogate of cerebral blood flow (CBF), with the assumption that FV is proportional to CBF if the unknown diameter of insonated vessel remains constant. Thanks to its non-invasive nature and bedside availability, TCD-FV may be widely used in neurocritical care units, but as for ICP, FV also seems to carry more information than its mean value. Many studies noted relevant information encoded in FV waveform changes, e.g., increased peak-systolic (FV_s_) and a drop in end-diastolic (FV_d_) flow velocity during periods of inadequate cerebral perfusion [11,12]. A ratio between those extreme points in FV waveform, normalised to FV mean value, is known as Pulsatility Index (PI), and was originally believed to be a descriptor of cerebrovascular resistance and a foundation for non-invasive ICP estimation [13]. However, despite extensive work carried out to build a non-invasive ICP estimator upon FV values, e.g., FV_d_, or FV-derived indices, such as PI, there is still no reliable alternative to a direct ICP measurement [14,15], leaving a need for further research into FV waveform analysis methods.

The aim of this work is to present a methodology for analysis of changes in shape of TCD pulse waveform course in relation to that of ABP. Such a method, if robust, may serve as an alternative of non-invasive plateau waves detection.

## 2. Materials and Methods

Retrospective analysis of previously published data was performed [5,16,17,18,19,20]. From among 160 head-injured patients treated in the Neurocritical Care Unit, Addenbrooke’s Hospital in the years 1992–1995, 31 presented ICP plateau waves, defined as a sudden rise in ICP above the value of 40 mmHg that lasted at least 5 min. All of the patients were analgosedated, intubated and mechanically ventilated to achieve mild hypocapnia (end-tidal pCO_2_ 3.5–4 kPa); further details on patients’ clinical data, treatment protocol and data acquisition methodology were thoroughly described elsewhere [5]. In 96 patients, intermittent recording of ICP, ABP and TCD was performed on a daily basis, for periods ranging from 20 min to 4 h. A total of 18 recordings captured in 8 patients (2 women, 6 men; age 19–36 years) covered the entire course of plateau ICP wave and, therefore, were available for analysis. Artefacts resulting from nursing interventions were removed manually.

Recordings were manually divided into periods of baseline ICP (normal or elevated) and stable phases of plateau (Figure 1). An automatic detection of points characteristic to each cardiac cycle, that is maximal (peak of systole) and minimal (end-diastolic) values [21], was performed on courses of raw ABP and FV signals (Figure 2). Subsequently, signals were sliced into single cardiac cycles, and then, the representative cycle of ABP as well as that of FV were derived by a process of aligning all individual cycles and averaging them, which reduced the noise. This technique also allowed us to reduce time delay between ABP and FV signals.

Representative cycles were normalised so as for ABP and FV to assume values between 0 and 1 (see Figure 3). Three variants of calculating dissimilarity between ABP and FV waveforms, the latter referenced as Difference Index (DI), were proposed:$${\mathrm{DI}}_{\mathrm{dif}}={\left\langle {\mathrm{ABP}}_{t}-{\mathrm{FV}}_{t}\right\rangle }_{t}={{\langle \mathrm{ABP}}_{t}\rangle }_{t}-{{\langle \mathrm{FV}}_{t}\rangle }_{t}\;,\;{\mathrm{DI}}_{\mathrm{abs}}={\left\langle \left|{\mathrm{ABP}}_{t}-{\mathrm{FV}}_{t}\right|\right\rangle }_{t}\;,\;{\mathrm{DI}}_{\mathrm{rsq}}=\sqrt{{\left\langle {\left({\mathrm{ABP}}_{t}-{\mathrm{FV}}_{t}\right)}^{2}\right\rangle }_{t}}\;,$$
where ABP_t_ and FV_t_ stand for normalised values of ABP and FV curves at time t, |·| symbolises the absolute value, and ${\left\langle \cdot \right\rangle }_{t}$ indicates averaging over time within the representative cycle. While calculating DI_dif_ depends on relation of the ABP and FV curves to each other, that is which curve is above the other in given time point, DI based on absolute value (DI_abs_) and root square (DI_rsq_) is proposed as independent from ABP to FV curve position influence. Note that averaging the differences between ABP and FV curves over time, i.e., ${\mathrm{DI}}_{\mathrm{dif}}$ variant, is equivalent to subtracting the average values of ABP and FV curves.

Pulsatility Index (PI) was calculated according to formula presented by Gosling [22]:$$\mathrm{PI}=\frac{{\mathrm{FV}}_{s}-{\mathrm{FV}}_{d}}{{\mathrm{FV}}_{m}}$$
where FV_s_ stands for peak-systolic, FV_d_—end-diastolic and FV_m_—mean value of FV. FV_s_ and FV_d_ were found as, respectively, maxima and minima of raw FV time course in 2 s long segments. PI values were first obtained in 10 s-long segments and then averaged over time to produce one representative value for each baseline and plateau episode.

Normal distribution of the data was confirmed with Shapiro–Wilk test and, therefore, a paired two-side *t*-test was used for further analysis. Values of DI before and during an ICP plateau wave were compared. DI ability to differentiate between baseline and plateau was compared against that of PI with receiver operating characteristic (ROC) curve analysis.

Intracranial pressure was monitored with intraparenchymal sensors, either the Camino Direct Pressure Monitor (Camino Laboratories, San Diego, CA, USA) or the Codman Microsensors ICP Transducer (Codman and Shurtleff, Inc., Raynham, MA, USA).

Signal recording, editing, and pulse analysis were performed in ICM+ (https://icmplus.neurosurg.cam.ac.uk; Cambridge Enterprise Ltd., Cambridge, UK; accessed on 13 May 2021). Following calculations and statistical analysis were conducted in Matlab (MathWorks, Natick, MA, USA).

The data were collected during routine multimodal monitoring included in Protocol 30 REC 97/290, approved by a local bioethical committee. After anonymization, digital recordings were retrospectively analysed as a part of clinical audit.

## 3. Results

A total of 18 recordings of plateau waves preceded by stable baseline period were available for analysis. Mean values, standard deviation and results of paired *t*-test (baseline vs. plateau) are presented in Table 1. During ICP plateau wave, values of DI were significantly higher (*p* < 0.0001). Values of DI are different in three variants. According to ROC analysis presented in Table 2 and Figure 4, DI_dif_ yields somewhat higher ability to detect plateau waves as opposed to PI; however, this difference is not significant. Amongst analysed indices, DI_dif_ provides highest accuracy (0.875) and area under a curve (0.918).

## 4. Discussion

In pursuit of a reliable, repeatable and user-independent non-invasive ICP estimator, many modalities using various signal sources have been introduced, e.g., assessment of ICP transmission through cochlear aqueduct or into the optic nerve sheath, measurement of sound attenuation within the brain parenchyma, venous opthtalmodynamometry, and others, including minimally invasive epidural sensors [23]. Among numerous techniques available, TCD-waveform-based methods present some significant advantages, especially a possibility of continuous monitoring that allows for identifying both long-term trends and rapid changes of vascular origin.

Introduction of DI is an attempt to use simple, graphically-derived estimator to assess exceptionally complex intracranial haemodynamics. The DI concept was based on the observation that in physiological conditions, there is a striking resemblance between ABP and FV waveforms. They might be roughly approximated to each other, since ABP is a major driving force for CBF and, concordantly, FV. The more pathological intracranial hemodynamics are, the more divergent ABP and FV waveform courses become.

Calculation of DI can be summarised in three steps: 1. fragmentation of ABP and FV signal according to cardiac cycle; 2. obtaining the normalised representative ABP and FV cycles by a process of aligning all individual cycles and averaging them; 3. applying the formula to obtain the Difference Index, which might be described by the area between the curves. In analysed clinical data, calculated difference between ABP and FV courses is significantly higher (*p* < 0.0001) during plateau wave.

PI was chosen as reference to DI because during plateau wave, PI value also differed significantly (*p* < 0.0001). Despite not being a good descriptor of ICP, in limited clinical situations, that is normal CO_2_ pressure and arterial pressure, PI may serve as an indicator of severe cerebral perfusion pressure (CPP) impairment.

DI, as a parameter derived from entire cardiac cycle waveform, provides a different angle on phenomena previously analysed by PI as single points: increased peak-systolic and diminished end-diastolic FV. Whereas PI is focused on the ratio between those extreme values, DI underlines diminished flow during the entire diastolic phase of the cardiac cycle. A drop in end-diastolic FV (FV_d_) value during the plateau wave was previously linked with vasocollapse due to CPP below the threshold of critical closing pressure [17]. Diminished FV_d_, either alone or combined with PI, is a valuable indicator of ischemic brain insult following traumatic brain injury [11,12,24]. The ratio FV/FV_d_ can be utilised in non-invasive ICP estimation [12,15].

Values of DI_dif_ can be negative and are lower than those of DI_abs_ and DI_rsq_ (Table 1). This is because for DI_dif_, a positive difference (ABP*_t_* > FV*_t_*) in one time point can be neutralised by a negative difference (ABP*_t_* < FV*_t_*) in another point. On the other hand, DI_abs_ and DI_rsq_ treat positive as well as negative differences in the same way regardless of their sign, and simply accumulate them. DI_rsq_ scores large differences between ABP and FV slightly stronger than DI_abs_ does. In practice, however, both indices produce very similar results (*R* = 0.96). DI_dif_ is also correlated with DI_abs_ (*R* = 0.57) and DI_rsq_ (*R* = 0.49).

According to ROC curves’ analysis, DI_dif_ yields higher strength in plateau wave detection than PI (AUC 0.918 vs. 0.898, respectively); however, this improvement is not statistically significant in the study sample. Dissimilarities in calculation methodology of DI and PI resulted in differences in sensitivity (0.813 vs. 0.688) and specificity (0.938 vs. 1, respectively). In the clinical setting, remarkably higher sensitivity of DI_dif_ to a potentially harmful intracranial hypertension appears to be of value. Additionally, DI_dif_ provides a better trade-off between sensitivity and specificity, resulting in the highest probability of detecting the real state of ICP (accuracy = 0.875).

In an example of ICP plateau waves, the most straightforward explanation of observed ABP-FV waveform divergence would be a decrease in CPP caused by a sudden rise in ICP. However, numerous other factors shaping CBF have been identified. Some of them, such as heart rate and ABP pulse amplitude, can actually be measured; others might be only estimated in models, e.g., cerebrovascular resistance or arterial bed compliance. An ICP plateau wave, as a product of a vasodilatatory vicious circle, refers to a series of events affecting cerebral vasculature in many ways [25]. It is yet to be determined whether difference in ABP-FV waveforms reflects diminished CPP, elevated ICP, low cerebrovascular resistance, depleted cerebrospinal compliance or state of impaired cerebral autoregulation. If further tested and interpreted in a context of aforementioned parameters, DI might be found to be useful in the setting of invasive ICP measurement as an aid providing a physiological background of observed changes.

Additionally, alterations in PaCO_2_ are known to change cerebrovascular resistance and, therefore, can influence FV-based calculations. In our cohort, patients were mechanically ventilated to achieve mild hypocapnia and alterations in PaCO_2_ were not recorded before and after an onset of plateau wave. Therefore, despite that conclusions cannot be drawn from presented material, we realise that an assessment of the relationship between CO_2_ and DI would provide valuable information about specificity of the presented index and remains a goal for our further work.

ABP serves as an auto-reference for measurements of FV change, which is a “true” variable. During plateau wave, changes in circulatory system have traditionally been considered minor (in our data, there was no significant difference in mean ABP values, *p* = 0.9). However, recent research indicates significance of sympathetic activity affecting circulation, such as baroreflex and heart rate variability [26]. Using ABP curve as a reference makes DI potentially resistant to changes affecting primarily the cardiac system.

Nevertheless, operations performed on ABP and FV waveforms may result in their distortion. In order to refer FV to ABP—different physical quantities, i.e., velocity and pressure, whose values are expressed in different units—a process of normalisation was necessary. Minimal and maximal values in both curves were considered as reference points, and hence, the normalisation was performed in such a way that they assumed values 0 and 1, respectively. However, peak-systolic and end-diastolic FV do change significantly during a plateau wave when compared with baseline (*p* ≈ 10^−4^), and our reference points are shifted as well, which makes calculation of DI potentially less robust. Elimination of that problem could potentially strengthen DI. It is difficult, however, to propose an alternative definition of reference point that would be valid for both ABP and TCD-FV waveforms.

Although analysed material was recorded in the 1990s, in our opinion the age of the data is not a significant obstacle because, in a sense, the “old” data could provide better insight into plateau waves’ physiology than those available today. A classic, CPP-oriented treatment protocol that was used in fully sedated patients in those times consisted of only limited intervention when ICP raised above 25 mmHg: mainly infusion of intravenous mannitol solution. Thanks to digitalised, high-resolution TCD recordings, our retrospective data might be interpreted as a “natural course” of ICP plateau waves not affected with therapeutical interventions.

## 5. Limitations

Our results should be perceived as preliminary. Further technical improvement, validation in a whole-time series, in a larger cohort and in other clinical scenarios is necessary.

We used pressure measured in radial artery and zeroed at the level of the heart as a surrogate of cerebral blood pressure. The application of transformation approximating ascending aortic pressure could potentially influence DI.

DI is susceptible to limitations inherent to any TCD measurements, such as signal transmission attenuation through the cranial vault. With currently available TCD methodology the lack of “acoustic window” in the temporal squama (which affects up to 8% of the adult population) simply precludes the applicability of DI.

In order to detect plateau wave, a baseline recording is required. In the clinical setting, without invasive ICP measurement, it is difficult to assess whether ICP is elevated or not. Studies on healthy subjects would potentially provide reference values of DI.

Improvement of normalisation could strengthen power of DI in detecting plateau wave.

In this preliminary work, all 18 plateau waves identified in 8 patients were treated as unrelated events. More rigorous statistical analysis in a larger cohort, however, could take into account the issue of potential dependence among subjects’ multiple measurements by employing, e.g., a mixed-effects model.

## 6. Conclusions

DI is a novel modality for FV analysis with potential for real-time, non-invasive plateau wave detection. Further technical improvement and validation is necessary.

## Figures and Tables

**Figure 1 brainsci-11-01000-f001:**
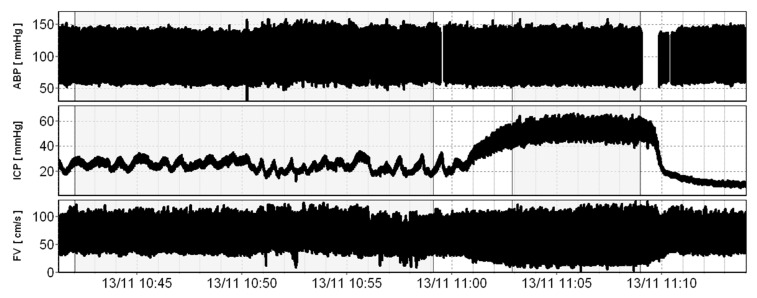
A recording of a plateau wave preceded with baseline. Analysed periods are shaded.

**Figure 2 brainsci-11-01000-f002:**
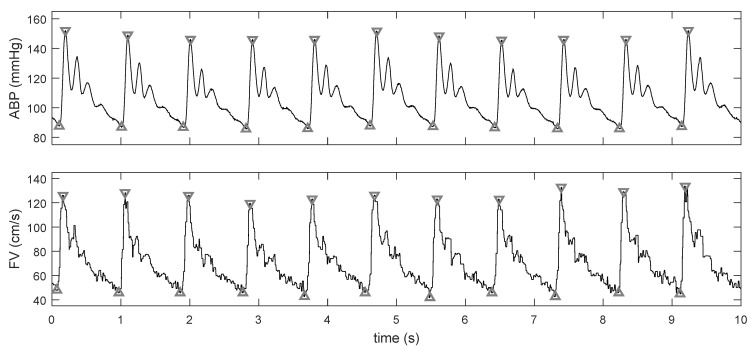
Detection of cardiac cycle: maximal (peak-systolic) and minimal (end-diastolic) values marked with triangles.

**Figure 3 brainsci-11-01000-f003:**
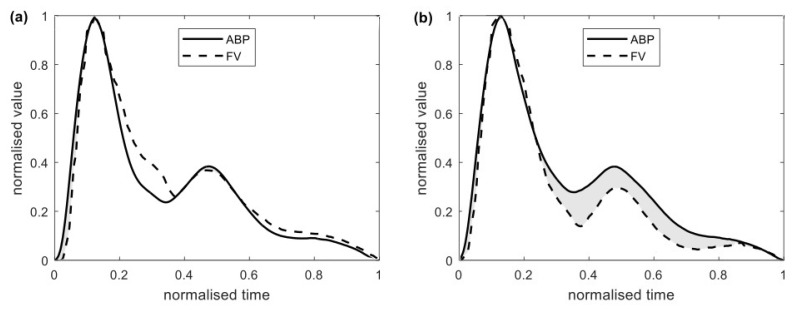
Representative cycles of ABP and FV waves at baseline (**a**) and during a plateau wave (**b**), obtained after the procedure of time alignment and normalisation described in the Materials and Methods section. The area where the normalised value of FV is lower than the one of ABP is shaded.

**Figure 4 brainsci-11-01000-f004:**
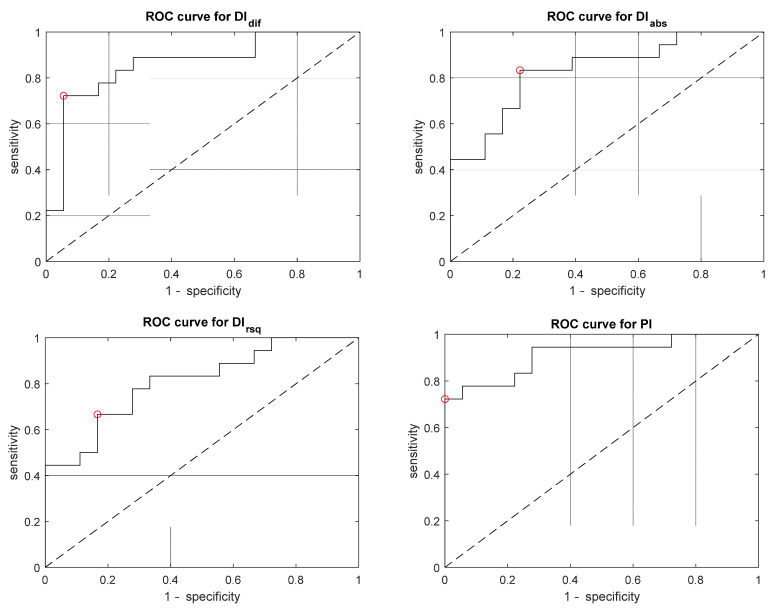
Receiver operating characteristic (ROC) curves showing the performance of given indices to discriminate between baseline ICP vs. plateau wave. The circle denotes the optimal operating point.

**Table 1 brainsci-11-01000-t001:** Basic characteristics of physiological parameters at baseline and during ICP plateau wave.

	Baseline (Mean ± Std. Dev.)	Plateau (Mean ± Std. Dev.)	*p* Value (Paired Two-Side *t*-Test)
ABP (mmHg)	95.1 ± 2.4	94.8 ± 2.3	0.9
ICP (mmHg)	24.9 ± 5.8	51.7 ± 2.3	0.0001
FV (cm/s)	61.1 ± 2.4	48.6 ± 2.2	0.006
FV_s_ (cm/s)	116.0 ± 4.1	127.3 ± 5.3	0.0003
FV_d_ (cm/s)	36.07 ± 2.3	21.25 ± 2.01	<10^−4^
PI	1.454 ± 0.094	2.446 ± 0.142	<10^−4^
DI_dif_	0.006 ± 0.031	0.054 ± 0.032	<10^−4^
DI_abs_	0.045 ± 0.021	0.082 ± 0.029	<10^−4^
DI_rsq_	0.061 ± 0.027	0.104 ± 0.037	<10^−4^

ABP—arterial blood pressure, ICP—intracranial pressure, FV—flow velocity, FV_s_—peak-systolic FV, FV_d_—end-diastolic FV, PI—Gosling pulsatility index, DI_dif_, DI_abs_ and DI_rsq_—three variants of the Difference Index as defined in the Materials and Methods section.

**Table 2 brainsci-11-01000-t002:** Receiver operating characteristic (ROC) curve showing the performance of given indices to discriminate between baseline ICP vs. plateau wave. AUC—area under the curve, CI—confidence interval.

	DI_dif	DI_abs	DI_rsq	PI
Sensitivity	0.813	0.813	0.813	0.688
Specificity	0.938	0.750	0.750	1
AUC	0.918	0.813	0.797	0.898
AUC 95% CI low	0.733	0.615	0.592	0.723
AUC 95% CI up	0.987	0.929	0.922	0.976
Accuracy	0.875	0.781	0.781	0.844

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
