# Peer review of "Change in Blood Flow Velocity Pulse Waveform during Plateau Waves of Intracranial Pressure"

_brainsci, 2021, doi:10.3390/brainsci11081000_

Round 1

Reviewer 1 Report

In this well-written manuscript, a retrospective analysis as to the noninvasive assessment of intracranial pressure (ICP) based on formerly published data from a cohort of head injured patients is presented. The reviewer suggest a major revision of this work. The revision should address the following issues:

As compared to several invasive methods, the advantage of proper noninvasive assessment of ICP will always be that it is inherently less risky regarding iatrogenic brain injury. Therefore, other ways of less invasive and noninvasive assessment of ICP (e.g., placement of sensors epidurally, acoustic methods, otic methods, venous ophthalmodynamometry, other optic nerve sheath diameter-related methods) have been proposed, and the respective references deserve citations in the introduction or discussion section of this manuscript.

Obtaining informed consent for the analysis of data recorded routinely at the bedside some 25-30 years ago appears neither necessary nor feasible, and the authors have already made the appropriate statement at the end of their manuscript. However, a (short) statement as to the approval of this study by an ethics committee is missing.

The reviewer understands that the data in this study originates from a patient cohort treated in the early 1990s. It would, however, be highly interesting to know if changes in relevant clinical parameters (e.g., pupillary function; onset of posturing; Glasgow Coma E, M, and V scores; depth of sedation) before and after the onset of the plateau waves coincided with the observed changes in the Difference Indexes.

One limitation of the approach as proposed here is that not in all patients, the windows in the temporal bone allow transcranial doppler sonography. This needs to be discussed.

Author Response

We would like to thank the Reviewers for time spent on careful reading and their expert academic support towards improving our manuscript

Point 1: As compared to several invasive methods, the advantage of proper noninvasive assessment of ICP will always be that it is inherently less risky regarding iatrogenic brain injury. Therefore, other ways of less invasive and noninvasive assessment of ICP (e.g., placement of sensors epidurally, acoustic methods, optic methods, venous ophthalmodynamometry, other optic nerve sheath diameter-related methods) have been proposed, and the respective references deserve citations in the introduction or discussion section of this manuscript.

Response 1: A new paragraph with a reference to a comprehensive review article concerning mentioned less-invasive and non-invasive methods of ICP monitoring has been added to Discussion (page 5/6, line 157-164)

Point 2: Obtaining informed consent for the analysis of data recorded routinely at the bedside some 25-30 years ago appears neither necessary nor feasible, and the authors have already made the appropriate statement at the end of their manuscript. However, a (short) statement as to the approval of this study by an ethics committee is missing.

Response2: We would like to thank the Reviewer for highlighting this important issue. A statement pertaining to the ethics committee approval has been added in the Methods section (page 3, line 122-124).

Point 3: The reviewer understands that the data in this study originates from a patient cohort treated in the early 1990s. It would, however, be highly interesting to know if changes in relevant clinical parameters (e.g., pupillary function; onset of posturing; Glasgow Coma E, M, and V scores; depth of sedation) before and after the onset of the plateau waves coincided with the observed changes in the Difference Indexes.

Response 3: We do agree with the Reviewer that clinical manifestation of plateau waves would be of interest for the reader of our report. Following his/her suggestion we re-examined all available clinical records of these patients for any traceable clinical features which could potentially coincide with plateau waves. Although clinical data of all studied patients have been referred to a former publication of Czosnyka et al. (Reference 5), we have supplemented the Materials and Methods section with information that all the patients were intubated, sedated and mechanically ventilated (page 2, lines 78-80). Unfortunately, no recorded clinical observations relating to pupil diameter during ICP elevations have been available in the relevant medical documentation.

Point 4: One limitation of the approach as proposed here is that not in all patients, the windows in the temporal bone allow transcranial doppler sonography. This needs to be discussed.

Response 4: We thank the Reviewer for this apt comment indicating one more limitation of our DI proposition that undoubtedly deserves mentioning. The Limitations section has been supplemented with appropriate note (page 7, line 258-261). Of course, the only possible way of passing this obstacle by is future progress in transcranial ultrasound technique.

Reviewer 2 Report

I read the authors work entitled “Change in blood flow velocity pulse waveform during plateau waves of intracranial pressure” which demonstrates how differences between the arterial pulse waveform and intracranial pressure waveform may inform the presence of abnormal plateau waves. There is limited clinical utility given the necessity of performing a “baseline” measurement when baseline is unknown but physiologically this is an interest approach to understanding the relationship between ICP and ABP waveforms.

1) Could the authors provide a definition that was used to define “plateau” waves for this work? The example given lasts only about 10 minutes but the amplitude is quite high; in my experience, I have seen very similar physiology with lower-amplitude elevations in ICP; many last closer to 15-20 minutes however.

2) Could the authors comment on the age of the cohort – these appear to be patients monitored some 25 years ago which seems a bit odd. Is there any contemporaneous data that could be used or were there differences in the TCD technology at that time that the reader should be aware of?

3) In the results, the authors state that DIdif yields higher ability to detect plateau waves but in the discussion, it is acknowledged that these are not significant differences. This should be clarified in the Results so as to not skew interpretation.

4) In the Discussion could the authors comment if DI is impacted by CO2?

5) Could the FV/FVd be used instead of the complex calculations necessary for the DI? It might be useful to compare this value in terms of distinguishing stable vs. plateau ICP

6) Could the authors comment on the potential role of continuous or automated TCD and perhaps how this could be useful to define plateau waves physiologically? In those undergoing simultaneous ICP monitoring, would the DI add additional important information?

7) The authors nicely highlight that the ABP is recorded from the radial artery and levelled at the phlebostatic axis. I assume the ICP is being measured from an external ventricular drainage catheter levelled at the tragus, however I didn’t see this explicitly stated.

Author Response

We would like to thank the Reviewer for time spent on careful reading and expert academic support towards improving our manuscript.

Point 1: Could the authors provide a definition that was used to define “plateau” waves for this work? The example given lasts only about 10 minutes but the amplitude is quite high; in my experience, I have seen very similar physiology with lower-amplitude elevations in ICP; many last closer to 15-20 minutes however.

Response 1: We thank the Reviewer for this comment of fundamental meaning for the readers of our work. We added a definition to the Materials and Methods (page 2, line 77-78).

Point 2: Could the authors comment on the age of the cohort – these appear to be patients monitored some 25 years ago which seems a bit odd. Is there any contemporaneous data that could be used or were there differences in the TCD technology at that time that the reader should be aware of?

Response 2: We thank the Reviewer for this comment. Indeed, our data may seem outdated and color Doppler was since this time introduced to obtain more accurate blood flow velocities in the middle cerebral artery thanks to the correction of the angle of insonation. Nevertheless, the “blind” (and not color) method is still used to continuous Doppler monitoring of the MCA. Additionally, we believe that our work in a sense actually benefits from technically perfect data, but recorded before more intense interventions were introduced to treat raises in ICP. As an explanation of our premise, we added a sub-paragraph to the Discussion (page 7, line 243-250).

Point 3: In the results, the authors state that DIdif yields higher ability to detect plateau waves but in the discussion, it is acknowledged that these are not significant differences. This should be clarified in the Results so as to not skew interpretation.

Response 3: The Results section has been changed to clarify that important point (page 3, line 137-138).

Point 4: In the Discussion could the authors comment if DI is impacted by CO2?

Response 4: We have expanded the Discussion section (page 7, line 217-223). The information about mechanical ventilation goals, however cited in Reference 5, has also been highlighted in Materials and methods (page 2, line 79-80).

Point 5: Could the FV/FVd be used instead of the complex calculations necessary for the DI? It might be useful to compare this value in terms of distinguishing stable vs. plateau ICP

Response 5: It is an apt comment, indicating the Reviewer’s deep insight into the context of our manuscript. We had calculated and tested the potential of FV/FVd index in differentiating periods before and during ICP plateau wave, even though former works (Cardim et al. Reference 15) indicated its inferiority in a task of detection of ICP>20 mmHg. Also in our preliminary comparison, this index showed inferiority in a comparison to DI. For this reason, we decided not to introduce this indicator to our estimation of DI.

Point 6: Could the authors comment on the potential role of continuous or automated TCD and perhaps how this could be useful to define plateau waves physiologically? In those undergoing simultaneous ICP monitoring, would the DI add additional important information?

Response 6: We thank the Reviewer for these pertinent remarks. As to the potential role of continuous or automated TCD, its greatest value is an ability of identification long-term trends and rapid changes of vascular origin, also in patients subjected to direct ICP measurement and such comment has been added to manuscript (Discussion section, page 5/6, lines: 161-164)

As a response to the question if DI is able to bring in an additional important information, especially this pertaining to physiological background of plateau waves generation, we can offer only a kind of speculation, at least at the present stage of our research. Nevertheless, we would like to attract the Reviewer’s attention at a paragraph of discussion (page 6, lines 198-205) where we have listed all hitherto identified potential physiological mechanisms due to which BP wave envelope becomes divergent from FV waveform during plateau wave. To comply with the Reviewer’s comment, we have placed at the end of this paragraph a short note in which we suggest that further experience might help in ascribing DI to one or some of the listed physiological mechanism (page 6, line 214-216).

Point 7: The authors nicely highlight that the ABP is recorded from the radial artery and levelled at the phlebostatic axis. I assume the ICP is being measured from an external ventricular drainage catheter levelled at the tragus, however I didn’t see this explicitly stated.

Response 7: ICP has been measured with intraparenchymal sensors, either the Camino Direct Pressure Monitor or the Codman Microsensors ICP Transducer (Reference 5). We added this information to Materials and Methods section (page 3, line 122-124).

Round 2

Reviewer 1 Report

The reviewer's comments have been adequately addressed.

Reviewer 2 Report

The authors have responded to all of my comments. The manuscript reads well and I have no major or minor concerns that require revision at this time.